# AREG Upregulation in Cancer Cells via Direct Interaction with Cancer-Associated Fibroblasts Promotes Esophageal Squamous Cell Carcinoma Progression Through EGFR-Erk/p38 MAPK Signaling

**DOI:** 10.3390/cells13201733

**Published:** 2024-10-19

**Authors:** Takashi Nakanishi, Yu-ichiro Koma, Shoji Miyako, Rikuya Torigoe, Hiroki Yokoo, Masaki Omori, Keitaro Yamanaka, Nobuaki Ishihara, Shuichi Tsukamoto, Takayuki Kodama, Mari Nishio, Manabu Shigeoka, Hiroshi Yokozaki, Yoshihiro Kakeji

**Affiliations:** 1Division of Pathology, Department of Pathology, Kobe University Graduate School of Medicine, Kobe 650-0017, Japan; 215m857m@stu.kobe-u.ac.jp (T.N.); shoji224@med.kobe-u.ac.jp (S.M.); rikuarw@med.kobe-u.ac.jp (R.T.); h450@med.kobe-u.ac.jp (H.Y.); 239m863m@stu.kobe-u.ac.jp (M.O.); 213m889m@gsuite.kobe-u.ac.jp (K.Y.); n.ishihara@people.kobe-u.ac.jp (N.I.); stsuka@med.kobe-u.ac.jp (S.T.); takodama@med.kobe-u.ac.jp (T.K.); marin@med.kobe-u.ac.jp (M.N.); mshige@med.kobe-u.ac.jp (M.S.); hyoko@med.kobe-u.ac.jp (H.Y.); 2Division of Gastro-Intestinal Surgery, Department of Surgery, Kobe University Graduate School of Medicine, Kobe 650-0017, Japan; kakeji@med.kobe-u.ac.jp; 3Division of Hepato-Biliary-Pancreatic Surgery, Department of Surgery, Kobe University Graduate School of Medicine, Kobe 650-0017, Japan; 4Division of Obstetrics and Gynecology, Department of Surgery Related, Kobe University Graduate School of Medicine, Kobe 650-0017, Japan

**Keywords:** esophageal squamous cell carcinoma (ESCC), cancer-associated fibroblasts (CAFs), direct co-culture, amphiregulin (AREG), epidermal growth factor receptor (EGFR)

## Abstract

Cancer-associated fibroblasts (CAFs) are a key component of the tumor microenvironment and significantly contribute to the progression of various cancers, including esophageal squamous cell carcinoma (ESCC). Our previous study established a direct co-culture system of human bone marrow-derived mesenchymal stem cells (progenitors of CAFs) and ESCC cell lines, which facilitates the generation of CAF-like cells and enhances malignancy in ESCC cells. In this study, we further elucidated the mechanism by which CAFs promote ESCC progression using cDNA microarray analysis of monocultured ESCC cells and those co-cultured with CAFs. We observed an increase in the expression and secretion of amphiregulin (AREG) and the expression and phosphorylation of its receptor EGFR in co-cultured ESCC cells. Moreover, AREG treatment of ESCC cells enhanced their survival and migration via the EGFR-Erk/p38 MAPK signaling pathway. Immunohistochemical analysis of human ESCC tissues showed a positive correlation between the intensity of AREG expression at the tumor-invasive front and the expression level of the CAF marker FAP. Bioinformatics analysis confirmed significant upregulation of *AREG* in ESCC compared with normal tissues. These findings suggest that AREG plays a crucial role in CAF-mediated ESCC progression and could be a novel therapeutic target for ESCC.

## 1. Introduction

Esophageal cancer is the seventh leading cause of cancer death worldwide, with 445,129 deaths in 2022, accounting for 4.6% of all cancer-related deaths [1]. It is classified into two main histological subtypes: esophageal adenocarcinoma (EAC) and esophageal squamous cell carcinoma (ESCC). In 2020, ESCC accounted for approximately 85% of all new esophageal cancer cases. However, ESCC has a lower survival rate than EAC, and deaths from ESCC are expected to continue rising [2]. Early diagnosis of ESCC is challenging, and surgical resection is often difficult due to active local invasion and lymph node metastasis. Additionally, ESCC often develops resistance to chemotherapy and radiation therapy, contributing to its poor prognosis [3]. Therefore, elucidating the mechanisms involved in the progression of ESCC is urgently needed.

Cancer-associated fibroblasts (CAFs) are a major component of the tumor microenvironment and play a critical role in tumor progression. Numerous studies have investigated the specific mechanisms by which CAFs promote tumor progression. CAFs are reported to originate from various cell types, including pancreatic and hepatic stellate cells, resident fibroblasts, mesenchymal stem cells (MSCs), adipocytes, adipose-derived MSCs, mesothelial cells, endothelial cells, myeloid cells, pericytes, epithelial cells, hematopoietic stem cells, and circulating bone marrow cells known as fibrocytes [4]. CAFs remodel the extracellular matrix (ECM), leading to tissue stiffening, which induces signaling that promotes cancer cell survival and growth [5]. Additionally, increased mechanical stress from tissue stiffening compresses blood vessels, leading to hypoxia and promoting more aggressive cancer phenotypes [6,7]. CAFs also secrete various matrix metalloproteases and other proteases that degrade the ECM, facilitating tumor invasion and metastasis [8]. Furthermore, CAFs produce a variety of cytokines, growth factors, chemokines, and exosomes that influence tumor growth and immune cell mobilization [4,9].

In previous studies, we established an indirect co-culture system between ESCC cells and bone marrow-derived MSCs, one of the progenitors of CAFs, to elucidate the mechanisms by which CAFs promote ESCC progression [10,11,12]. Recently, to further unravel the role of CAFs in a more physiologically relevant environment, we established a direct co-culture system between ESCC cells and MSCs and reported that *POSTN*, a gene upregulated in CAFs, contributes to ESCC progression [13]. The MSCs after direct co-culture exhibited high expression of the CAF marker *FAP* and the CAF-highly expressed genes *IL-6* and *MT2A*, demonstrating the significant utility of this direct co-culture model [13]. CAF-conditioned medium induces HOXA9 expression in epithelial ovarian cancer cells, leading to tumor-derived TGF-β2 induction and differentiation of MSCs into CAFs, promoting tumor growth [14]. Angiopoietin-like protein 3, which is upregulated in oral squamous cell carcinoma (OSCC) cells co-cultured with CAFs compared to monocultured OSCC cells, may be involved in OSCC progression by inducing the differentiation of MSCs into CAFs [15]. However, no studies to date have analyzed genes upregulated in ESCC cells following co-culture with CAFs.

In this study, we focused on genes upregulated in ESCC cells after direct co-culture with MSCs. We performed a cDNA microarray analysis between ESCC cells monocultured and co-cultured with MSCs. The results revealed a marked upregulation of amphiregulin (AREG) in co-cultured ESCC cells. AREG, a ligand for the epidermal growth factor receptor (EGFR), is known to play a critical role in the development and progression of various cancers, as well as in the resistance to radiation and chemotherapy [16]. In this study, we investigated the role of AREG in ESCC progression and its effects on CAFs.

## 2. Materials and Methods

### 2.1. Cell Lines and Cell Cultures

This study employed three human ESCC cell lines (TE-9, -10, and -15) obtained from the RIKEN BioResource Center (Tsukuba, Japan). To use cell lines of different levels of differentiation, TE-9 cells, which are poorly differentiated ESCC cells, and TE-10 and TE-15, which are highly differentiated ESCC cells, were selected [17]. These cell lines were maintained in Roswell Park Memorial Institute Medium (RPMI)-1640 medium (FUJIFILM Wako Pure Chemical Corporation, Osaka, Japan) supplemented with 10% fetal bovine serum (FBS; Sigma-Aldrich, St. Louis, MO, USA) and 1% penicillin–streptomycin–amphotericin B suspension (FUJIFILM Wako Pure Chemical Corporation) in a humidified incubator at 37 °C with 5% CO_2_. Human bone marrow-derived MSCs were purchased from the American Type Culture Collection (PCS-500-012; Manassas, VA, USA) and cultured in low-glucose Dulbecco’s Modified Eagle’s Medium (DMEM; FUJIFILM Wako Pure Chemical Corporation) supplemented with 10% FBS and 1% penicillin–streptomycin–amphotericin B suspension in a humidified incubator at 37 °C with 5% CO_2_.

### 2.2. Direct Co-Culture System

ESCC cells (TE-9, -10, and -15) were directly co-cultured with MSCs using a previously established method [13]. In brief, MSCs were seeded at a density of 3 × 10^5^ cells per 100 mm dish and cultured for 3 h in high-glucose DMEM supplemented with 10% FBS and 1% penicillin–streptomycin–amphotericin B suspension. Subsequently, ESCC cells were seeded at a density of 2 × 10^5^ cells per dish and co-cultured with MSCs for 4 days. Monocultures of both cell types cultured under similar culture conditions served as controls. Following co-culture or monoculture, cells were washed with phosphate-buffered saline (PBS; FUJIFILM Wako Pure Chemical Corporation) and detached from the dish using trypsin-ethylenediaminetetraacetic acid (EDTA) (FUJIFILM Wako Pure Chemical Corporation). The harvested cells were then mixed with anti-CD326 (epithelial cell adhesion molecule; EpCAM) microbeads (130-061-101; Miltenyi Biotec, Bergisch Gladbach, Germany) and subjected to magnetic separation using the autoMACS Pro Separator (Miltenyi Biotec) to isolate high-purity tumor cells.

### 2.3. cDNA Microarray Analysis

cDNA microarray analysis was performed on TE-9 monoculture (TE-9 mono) and TE-9 co-culture (TE-9 co) cells using the 3D-Gene Human Oligo Chip 25k (Toray Industries, Tokyo, Japan). The data from this analysis were deposited in the Gene Expression Omnibus (GEO) database under accession number GSE274064. In addition, a previous study’s cDNA microarray analysis (GSE244020) [13] comparing MSC monoculture and CAF-like cells (CAF9, MSCs directly co-cultured with TE-9) was also included in the present analysis.

### 2.4. qRT-PCR

RNA was isolated from cultured cells using the RNeasy Mini Kit (Qiagen, Hilden, Germany) following the manufacturer’s protocol. RNA concentration was then assessed using a NanoDrop Lite spectrophotometer (Thermo Fisher Scientific, Waltham, MA, USA). qRT-PCR was performed on the Applied Biosystems StepOne Real-Time PCR System (Applied Biosystems, Foster City, CA, USA) to quantify the expression levels of *AREG*, *EGFR*, and *GAPDH* using SYBR Green Master Mix and of *FAP*, *ACTA2*, and *ACTB* using TaqMan Gene Expression Master Mix (Applied Biosystems). The comparative Ct method was employed for data analysis. Ct values were normalized to a reference gene (*GAPDH* for SYBR Green and *ACTB* for Taqman) following the manufacturer’s instructions. Primer sequences for the SYBR Green qRT-PCR are as follows: *AREG*, 5′-TGAGATGTCTTCAGGGAGTG-3′ (forward) and 5′-AGCCAGGTATTTGTGGTTCG-3′ (reverse); *EGFR*, 5′-GAGAGGAGAACTGCCAGAA-3′ (forward) and 5′-GTAGCATTTATGGAGAGTG-3′ (reverse); *IL-6*, 5′-AATAACCACCCCTGACCCAAC-3′ (forward) and 5′-AATCTGAGGTGCCCATGCTAC-3′ (reverse); and *GAPDH*, 5′-GCACCGTCAAGCCTGAGAAT-3′ (forward) and 5′-ATGGTGGTCAAGACGCCAGT-3′ (reverse). Primer probes for the Taqman qRT-PCR are as follows: *FAP*, Hs00990806_m1; *ACTA2*, Hs00426835_g1; and *ACTB*, Hs01060665_g1.

### 2.5. Western Blot Analysis

Protein extraction was performed on ice using a lysis buffer containing: 50 mM Tris-HCl (pH 7.5), 125 mM NaCl, 5 mM EDTA, 0.1% Triton X-100, and 1% each of a protease and a phosphatase inhibitor cocktail (Sigma-Aldrich). A NanoDrop Lite spectrophotometer was employed to measure the protein concentration. SDS-PAGE was performed on 5–20% gradient gels (FUJIFILM Wako Pure Chemical Corporation) to separate proteins for molecular weight. Following electrophoresis, protein transfer to polyvinylidene difluoride membranes was performed using an iBlot2 system (Invitrogen, Carlsbad, CA, USA). Membranes were blocked with 5% skim milk in Tris-buffered saline with Tween 20 (TBS-T) for 30 min at 25 °C. Primary antibody incubation was performed overnight at 4 °C. After washing the membranes with TBS-T, the membranes were incubated with secondary antibodies for 90 min at 25 °C. ImmunoStar reagents (FUJIFILM Wako Pure Chemical Corporation) were used for chemiluminescent detection of protein bands, visualized using an ImageQuant LAS4000 mini (FUJIFILM, Tokyo, Japan). Due to the absence of chemiluminescence in the tri-color prestained protein markers (WIDE-VIEW TM Prestained Protein Size Marker III #234-02464; FUJIFILM Wako Pure Chemical Corporation), the markers are not visible in the raw data. The intensities of the protein bands were quantified using the ImageJ software version 1.53k (National Institutes of Health, Bethesda, MD, USA). The primary antibodies used for blotting are as follows: rabbit phosphorylated (p) Erk1/2 antibody (#9101, Cell Signaling Technology; CST, Danvers, MA, USA), rabbit Erk1/2 antibody (#9102, CST), rabbit pp38 MAPK antibody (#9211, CST), rabbit p38 MAPK (#9212, CST), rabbit pEGF Receptor (Tyr1068) antibody (#2234, CST), rabbit EGF Receptor antibody (#4267, CST), sheep FAP antibody (#AF3715, R&D Systems; Minneapolis, MN, USA), rabbit IL-6 antibody (#ab6672, Abcam, Cambridge, UK), rabbit αSMA antibody (#ab5694, Abcam), and rabbit β-actin antibody (#4970, CST). The secondary antibodies used are as follows: horseradish peroxidase (HRP)-conjugated donkey anti-rabbit IgG (#NA934V; Cytiva, Marlborough, MA, USA); and HRP-conjugated donkey anti-sheep IgG (#ab6900, Abcam).

### 2.6. Enzyme-Linked Immunosorbent Assay (ELISA)

Monocultured or co-cultured ESCC cells were seeded in 6-well plates at 2 × 10^5^ cells per well and cultured for 48 h in 3 mL of serum-free low-glucose DMEM. Subsequently, the supernatants were collected and analyzed for AREG protein concentration using a human amphiregulin Quantikine ELISA kit (#DAR00, R&D Systems) following the manufacturer’s protocol. The optical density of each well was measured at 450 nm and 570 nm using an Infinite 200 PRO microplate reader (Tecan, Mannedorf, Switzerland). AREG concentration in each well was calculated based on the absorbance values obtained from a standard curve.

### 2.7. Cell Survival and Growth Assay

For the cell survival assay, 1 × 10^4^ ESCC cells per well were seeded in serum-free RPMI-1640, while for the cell growth assay, 5 × 10^3^ ESCC cells per well were seeded in 1% FBS-supplemented RPMI-1640 in 96-well plates. Recombinant human amphiregulin protein (rhAREG, #262-AR, R&D Systems) (100 ng/mL) was added to the ESCC cells in some experiments. Additionally, in certain experiments, ESCC cells were pretreated for 24 h with either AG1478 (#S2728, Selleckchem, Houston, TX, USA) (10 μM) or dimethyl sulfoxide (DMSO, FUJIFILM Wako Pure Chemical Corporation) as a control. After 48 h, cell survival and growth were assessed by the dimethylthiazol-carboxymethoxyphenyl-sulfophenyl-tetrazolium (MTS) assay using the CellTiter 96 AQueous One Solution Reagent (Promega, Madison, WI, USA). The absorbance measurement at 492 nm was performed on the Infinite 200 PRO microplate reader. Similar experiments were performed on MSCs using serum-free low-glucose DMEM for the survival assay (3 × 10^3^ cells per well) and 1% FBS-supplemented low-glucose DMEM for the growth assay (1.5 × 10^3^ cells per well).

### 2.8. Transwell Migration Assay

For the migration assay, 1 × 10^5^ ESCC cells per well were seeded in 300 μL of serum-free RPMI-1640 on cell culture inserts (pore size 8.0 μm, BD Falcon, Lincoln Park, NY, USA) placed as the upper chambers of each well in 24-well plates. The lower chamber of the wells was filled with 800 μL of 0.1% FBS-supplemented RPMI-1640 and the upper chamber was brought into contact with the lower chamber. In some experiments, ESCC cells were treated with rhAREG (100 ng/mL) and pretreated with AG1478 (10 μM) or DMSO, similar to the MTS assay. After 48 h, cells that migrated to the underside of the membrane from the upper chambers were stained with Diff-Quik (Sysmex, Kobe, Japan), and the number of migrated cells was counted in five randomly selected fields. Similar experiments were performed on MSCs (2 × 10^5^ cells per well) using 0.1% and 1% FBS-supplemented low-glucose DMEM in the upper and lower chambers, respectively.

### 2.9. Wound Healing Assay

TE-9 and TE-10 cells were seeded at a density of 2 × 10^5^ cells per well in 24-well plates containing 600 μL of 10% FBS-supplemented RPMI-1640. After 24 h, upon reaching 100% confluence, a uniform scratch wound was created using a sterile 1000 μL pipette tip. The cells were washed with PBS, and four representative scratch fields per well were captured using a charge-coupled device camera (Olympus, Tokyo, Japan). Subsequently, the culture medium was replaced with serum-free RPMI-1640 with or without rhAREG (100 ng/mL), and the cells were incubated for an additional 24 h. After another PBS wash, the same fields from each well were re-imaged to assess cell migration. Wound closure was quantified using the Image J software.

### 2.10. Immunofluorescence

ESCC cells (TE-9, -10, and -15) were directly co-cultured with MSCs for 4 d, as described under “Direct Co-culture System”. After co-culture, cells were fixed with 4% paraformaldehyde for 10 min at 25 °C, then incubated overnight at 4 °C with primary antibodies against EpCAM (1:150; #2929; CST) and FAP (1:300; #AF3715; R&D Systems). After washing, cells were stained with DAPI (1:1000; FUJIFILM Wako Pure Chemical Corporation) and the following secondary antibodies: Cy3-conjugated anti-mouse IgG (1:200; Jackson ImmunoResearch Laboratories, West Grove, PA, USA) and Alexa Fluor-488-conjugated anti-sheep IgG (1:200; Jackson ImmunoResearch Laboratories). The slides were washed and visualized using a confocal laser-scanning microscope (LSM700; Carl Zeiss, Oberkochen, Germany). Images were analyzed with ZEN 2009 software version 5.5 SP1 (Carl Zeiss).

### 2.11. Tissue Samples

Tissue samples and clinical data from 68 patients who underwent surgical resection for ESCC at Kobe University Hospital from 2005 to 2010 were analyzed. Patients who had received preoperative chemotherapy and radiotherapy were excluded. Tissue samples were fixed in 10% formalin and paraffin-embedded. Histological and clinicopathological parameters were evaluated according to the 10th edition of the Japanese Classification of Esophageal Cancer [18,19] and the 7th edition of the Union for International Cancer Control TNM Classification of Malignant Tumours [20]. This study was conducted in accordance with the guidelines of the Declaration of Helsinki and approved by the Kobe University Institutional Review Board (approval no.: B210103 on 22 June 2021). The study adhered to pertinent ethical considerations, and all study participants provided informed consent.

### 2.12. Immunohistochemistry

Immunohistochemistry was performed on paraffin-embedded tissue sections (4 μm thick) using the Leica BOND-MAX automated system and the BOND Polymer Refine Detection Kit (Leica Biosystems, Nubloch, Germany). Mouse amphiregulin monoclonal antibody (1:2000, #66433-1-lg, Proteintech, Chicago, IL, USA) was used as the primary antibody. The staining intensity of AREG at the tumor-invasive front was compared with that of the normal esophageal squamous epithelium in the same section and was scored as follows: weak, 0; equivalent, 1; and strong, 2. Cases with scores of 0 and 1 were considered to have low AREG expression, and those with a score of 2 were considered to have high AREG expression. Immunohistochemical evaluations were independently performed on the patients’ clinical and pathology data by two pathologists (Y.-i.K. and H.Y.) and one surgeon (T.N.).

### 2.13. Bioinformatic Database Analysis

*AREG* and *EGFR* mRNA expression in ESCC tissues compared with normal tissues was analyzed using the TNMplot database “https://www.tnmplot.com/ (accessed on 17 September 2024)” [21]. The TNMplot database is a web application that enables real-time comparison of normal, tumor, and metastatic gene expression data by constructing an integrated database utilizing available transcriptome-level datasets. Furthermore, this database was employed to investigate the correlation between *AREG* and *FAP*, as well as *AREG* and *EGFR* gene expression in ESCC tissues, utilizing Spearman’s rank correlation analysis.

### 2.14. Statistical Analysis

All in vitro experiments were conducted in triplicate, and each experiment was independently replicated three times to ensure data reliability. For comparisons involving more than two groups, the data were presented as mean ± standard error of the mean and analyzed using the two-tailed Student’s t-test and Tukey–Kramer test. The relationship between the immunohistochemistry results and clinicopathological factors was assessed using the Chi-squared test. Survival curves for overall survival (OS), disease-free survival (DFS), and cancer-specific survival (CSS) were estimated using the Kaplan–Meier method. The log-rank test was employed to compare the survival difference between groups. Statistical analyses were conducted using IBM SPSS Statistics software version 22 (IBM Corp., Armonk, NY, USA) with a pre-defined significance level of *p* < 0.05.

## 3. Results

### 3.1. Direct Co-Culture with MSCs Increases AREG Gene Expression and Protein Secretion in ESCC Cells

A direct co-culture system for ESCC cell lines (TE-9, -10, and -15) and MSCs was established in a previous study to investigate the role of CAFs in the ESCC microenvironment (Figure 1A). The study demonstrated that MSCs acquired CAF-like properties after direct co-culture, and ESCC cells and MSCs after co-culture were referred to as TE co (TE-9, -10, and -15 co) and CAF-like cells (CAF9, 10, and 15), respectively. Similarly, ESCC cells and MSCs after monoculture were designated as TE mono (TE-9, -10, and -15 mono) and MSC mono, respectively. Double immunofluorescence staining revealed that EpCAM (red) was expressed in TE cells and FAP (green) was expressed in MSCs under direct co-culture conditions (Figure 1B), with direct contact between ESCC cells and CAFs was observed in certain regions. Furthermore, the previous study revealed that TE co exhibited enhanced malignant phenotypes, such as increased survival, growth, and migration, compared to TE mono. The previous study also performed a cDNA microarray analysis between MSC mono and CAF9 to identify genes highly expressed in CAFs. In contrast, in this study, the cDNA microarray analysis was performed between TE-9 mono and TE-9 co, aiming to identify genes highly expressed in the TE cells after co-culture with CAFs. Using a Venn diagram approach, genes that were highly expressed in CAFs were excluded, enabling us to focus on those upregulated in TE cells. This analysis identified 56 genes that were significantly upregulated in the cancer cells after direct co-culture (i.e., TE-9 co global normalization > 100 and TE-9 co/TE-9 mono ratio > 4). In contrast, 12,867 genes were found to have low expression in MSCs or CAF-like cells (i.e., MSC mono or CAF9 global normalization < 100), with 5 genes in common between them (Figure 1C). These five genes are shown in Table 1, with *amphiregulin* (*AREG*) exhibiting the highest fold change in TE-9 co. Therefore, we decided to focus on *AREG*. The qRT-PCR and ELISA analyses confirmed a significant increase in the mRNA expression and secreted protein level of AREG in TE-9, -10, and -15 co compared to TE-9, -10, and -15 mono, respectively (Figure 1D,E). These findings corroborated with the cDNA microarray results. Additionally, ELISA analysis of MSC mono and CAFs revealed that the secreted protein level of AREG was negligible in all cell types except CAF9. Even in CAF9, the secreted protein level of AREG was lower compared to that in all TE co (Appendix A).

### 3.2. AREG Secreted by ESCC Cells Promotes ESCC Cell Survival and Migration

To investigate the role of AREG in promoting malignant phenotypes of ESCC cells, we evaluated changes in the survival, growth, and migration of ESCC cells treated with rhAREG. The MTS assay revealed a dose-dependent increase in ESCC cell survival upon rhAREG treatment (Figure 1F), although no significant effect was observed on cell growth ability (Figure 1G). Similarly, the transwell migration assay demonstrated a dose-dependent enhancement in ESCC cell migration with rhAREG treatment (Figure 1H). The wound healing assay further confirmed the enhanced migration of TE-9 and TE-10 cells following AREG treatment (Appendix A); however, TE-15 was not tested as it was not suitable for this assay. Consequently, the transwell migration assay was employed in subsequent experiments to assess their migration ability.

### 3.3. EGFR-Erk/p38 MAPK Signaling Pathway Activation Mediates the Enhancement of ESCC Cell Survival, Growth, and Migration in Co-Cultures with CAFs

While the previous study demonstrated enhanced survival, growth, and migration of ESCC cells upon direct co-culture with MSCs, the role of AREG, a well-known EGFR ligand, in promoting these malignant phenotypes of ESCC cells remains unclear. To explore this, we focused on EGFR and phosphorylated EGFR (pEGFR) (Tyr1068), a reported marker of EGFR activation [22,23]. The cDNA microarray analysis (GSE274064) indicated an upregulation of *EGFR* expression in TE-9 co (TE-9 co/TE-9 mono ratio: 1.25), despite relatively low global normalization values (TE-9 mono: 28; TE-9 co: 35). qRT-PCR and Western blotting confirmed the upregulation of both mRNA and protein expression of EGFR in ESCC cells after co-culture (Figure 2A,B and Appendix A). Western blotting further revealed upregulation of pEGFR (Tyr1068) and pp38 MAPK protein expression in ESCC cells after co-culture, in addition to the previously reported increase in pErk (Figure 2B and Appendix A). To investigate the role of EGFR in the progression of malignant phenotypes of ESCC cells upon co-culture, ESCC cells were pretreated with an EGFR inhibitor AG1478 (10 μM) prior to co-culture. AG1478 treatment significantly abrogated the enhancement of ESCC cell survival, growth, and migration induced by co-culture (Figure 2C–E). Western blotting demonstrated sustained suppression of pEGFR (Tyr1068) protein expression in ESCC cells pretreated with AG1478 across all three ESCC cell lines (Figure 2F and Appendix A). Furthermore, Western blotting revealed that AG1478 pretreatment suppressed the co-culture-induced upregulation of pErk and pp38 MAPK protein expression in a subset of cell lines (pErk, TE-10 and -15; pp38 MAPK, TE-9 and -10) (Figure 2F and Appendix A). These findings suggested that co-culture with MSCs leads to an increase in EGFR expression and activation in ESCC cells, promoting cell survival, growth, and migration through the EGFR-Erk/p38 MAPK signaling pathway.

### 3.4. AREG Promotes ESCC Cell Survival and Migration Through the EGFR-Erk/p38 MAPK Signaling Pathway

ESCC cells treated with rhAREG for 96 h, mirroring the duration of the direct co-culture, exhibited significant upregulation of both mRNA and protein expression of EGFR, as well as the phosphorylation of EGFR (Tyr1068), Erk, and p38 MAPK as assessed by qRT-PCR and Western blotting (Figure 3A,B and Appendix A). To investigate the role of EGFR in the rhAREG-induced progression of malignant phenotypes of ESCC cells, ESCC cells were pretreated with AG1478 prior to rhAREG treatment. AG1478 pretreatment abrogated the rhAREG-induced enhancement of ESCC cell survival and migration (Figure 3C,D). Transient rhAREG treatment induced a rapid and significant upregulation of pEGFR (Tyr1068), pErk, and pp38 MAPK protein expression, as revealed by Western blotting (Figure 3E and Appendix A). Peak activation was observed at 10 to 30 min after rhAREG treatment. Notably, pretreatment with AG1478 abrogated these rhAREG-mediated effects. These findings suggested that AREG induces EGFR expression in ESCC cells and promotes ESCC cell survival and migration through the EGFR-Erk/p38 MAPK signaling pathway.

### 3.5. AREG Promotes Migration and CAF-like Differentiation of MSCs

The effects of AREG on MSCs were investigated. MSCs treated with rhAREG for 96 h showed significant upregulation of both mRNA and protein expression of CAF markers, FAP and IL-6, as assessed by qRT-PCR and Western blotting, although no change in αSMA expression was observed (Figure 4A,B and Appendix A). The MTS assay revealed no significant effect of rhAREG treatment on MSC survival and growth; however, the transwell migration assay revealed a marked increase in MSC migration following rhAREG treatment (Figure 4C–E).

### 3.6. AREG Expression Levels Are Significantly Elevated in ESCC Tissues and Positively Correlated with FAP Expression

Immunohistochemistry for AREG was performed on 68 human ESCC tissue samples to investigate the association between AREG expression and the patient’s prognosis and clinicopathological factors. AREG staining intensity was evaluated at the tumor-invasive front and compared with that of normal esophageal squamous epithelium. Patients were subsequently divided into low-intensity (*n* = 31) and high-intensity (*n* = 37) AREG expression (Figure 5A) based on histopathology scores. Kaplan–Meier analysis of OS, DFS, and CSS did not reveal significant differences (*p* = 0.640, *p* = 0.426, and *p* = 0.665, respectively) between the low-intensity and high-intensity groups (Figure 5B). However, the analysis showed that high-intensity *AREG* expression was positively correlated with *FAP* expression in ESCC tissues (Table 2). This association was also supported by the TNMplot database analysis (*p* < 0.001) (Figure 5C). The TNMplot database also revealed a significant upregulation of *AREG* expression in ESCC tissues compared with normal tissues (*p* < 0.01, R = 0.27) (Figure 5D). The TNMplot database analysis also revealed a positive correlation between *AREG* and *EGFR* expression in ESCC tissues (Appendix A). Additionally, *EGFR* expression was significantly upregulated in ESCC tissues compared with normal tissues (*p* < 0.01, R = 0.26) (Appendix A).

## 4. Discussion

Our previous studies have demonstrated that both indirect [10,11,12] and direct co-cultures [13] with CAFs promote malignant phenotypes of ESCC cells. Analysis of genes upregulated in CAFs after co-culture suggested the involvement of various paracrine factors influencing ESCC progression. Autocrine signaling in cancer cells has also been reported to play a crucial role in the progression of various cancers [24]. In this study, to investigate autocrine signaling in ESCC cells, we focused on *AREG*, a gene upregulated in ESCC cells following direct co-culture, and identified its potential role in ESCC progression.

AREG, a member of the EGF family, was first identified in the supernatants of phorbol myristate acetate-treated human breast cancer cells MCF-7 in 1988 [25]. It is constitutively expressed across various tissues, including the ovary, testis, mammary gland, placenta, pancreas, heart, colon, lung, spleen, and kidney [26]. It plays a crucial role in tissue development and organogenesis, especially in epithelial cells of the mammary gland [27], prostate [28], kidney [29], and lung [30]. AREG also regulates tissue homeostasis, as evidenced by reports suggesting that TLR4-mediated AREG expression contributes to the protection and repair of intestinal tissue [31]. Additionally, AREG is essential for skin homeostasis and acts as a potent stimulator of keratinocyte proliferation during wound healing [32]. The involvement of AREG in regulating inflammation and promoting tissue repair has been reported in diverse pathologies, including chronic respiratory diseases [33], liver fibrosis [34], and autoimmune diseases such as rheumatoid arthritis [35] and systemic lupus erythematosus [36]. Overexpression of AREG has been documented in various human cancer tissues, including those of the head and neck, breast, lung, liver, stomach, and colon [16]. The AREG autocrine loops play a significant role in tumor progression in specific cancers such as hepatocellular carcinoma [37], colorectal cancer [38], and ovarian cancer [39]. However, to the best of our knowledge, no prior studies have explored AREG expression and functional involvement in the initiation and progression of ESCC. This study aims to bridge this knowledge gap and presents the first investigation into the role of AREG in ESCC.

AREG is initially synthesized as a 252-amino acid transmembrane precursor, proAREG, which undergoes proteolytic cleavage or ectodomain shedding, similar to other EGF family members such as EGF and TGF-α, to release the soluble protein into the extracellular space [26]. This soluble AREG binds to and activates EGFR, initiating a cascade of signaling events that influence cellular behavior [26]. EGFR, a tyrosine kinase receptor, is a crucial component of fundamental signaling pathways in the cell and plays a pivotal role in tumor progression. The canonical mechanism of EGFR activation typically involves ligand binding to the extracellular domain of the receptor, promoting receptor dimerization [40]. This dimerization leads to the trans-autophosphorylation of key tyrosine residues in the receptor’s cytoplasmic tail by its kinase domain, triggering downstream signaling pathways, including Ras/MAPK, PI3K/AKT, PLCγ, and STAT [41,42].

Numerous studies have documented AREG-induced EGFR phosphorylation through this canonical mechanism. For instance, in pancreatic ductal adenocarcinoma (PDAC), TGF-β-induced AREG activates EGFR by phosphorylation at Tyr1068 and promotes PDAC metastasis [23]. Similarly, in colon cancer, overexpression of FGFR4 triggers AREG secretion, which promotes tumor growth through EGFR phosphorylation at Tyr1068 [22]. AREG secreted by mucoepidermoid carcinoma (MEC) cells may activate EGFR Tyr1068 in an autocrine manner, suggesting potential therapeutic applications for MEC [43]. Given these findings, we focused this study on EGFR phosphorylation at Tyr1068, a tyrosine residue highlighted in previous studies as a key phosphorylation site. We investigated the involvement of the AREG-EGFR axis in ESCC progression using a direct co-culture system, which we have previously reported [13]. ESCC cells exhibited increased pEGFR (Tyr1068) expression after co-culture. Experiments using AG1478, an EGFR tyrosine kinase inhibitor known to suppress pEGFR (Tyr1068) [44], demonstrated that the EGFR-Erk/p38 MAPK signaling pathway plays a critical role in the malignant phenotypes of ESCC cells. This study is the first to provide evidence of p38 MAPK pathway activation in ESCC cells following direct co-culture with MSCs. AREG treatment induced the phosphorylation of EGFR and activation of downstream Erk/p38 MAPK signaling, which were observed irrespective of treatment duration. Notably, we revealed not only increased EGFR phosphorylation but also elevated EGFR expression following direct co-culture and AREG treatment. This aligns with previous reports suggesting AREG promotes EGFR accumulation on the cell surface via receptor recycling [45,46].

CAFs are widely recognized for their crucial role in cancer progression within the tumor microenvironment of various cancers, including ESCC. Our laboratory has previously reported that the expression of the CAF marker FAP is associated with OS and DFS in patients with ESCC [10] and that factors secreted by CAFs, such as CCL2, IL-6 [10], PAI-1 [11], IGFBP2 [12], and POSTN [13], are involved in ESCC progression. Recent reports have suggested a link between AREG and CAFs. Lysophosphatidic acid-induced AREG secretion from CAFs promotes the invasiveness of breast cancer cells [47]. AREG secreted by myofibroblastic CAFs promotes the metastasis of PDAC [23]. In contrast to these reports, our study did not find increased *AREG* gene expression in CAFs. However, AREG secretion was enhanced in ESCC cells via co-culture with CAFs and was found to be involved in CAF-like differentiation and migration of MSC. These align with a study in which mice were co-injected with breast cancer cells and fibroblasts, demonstrating that AREG secreted by fibroblasts has both an autocrine effect that leads to fibroblast activation (increased αSMA expression) and a paracrine effect that protects cancer cells from apoptosis [48]. Recent studies have identified multiple CAF subtypes, with inflammatory CAF (iCAF) and myofibroblastic CAF (myCAF) being the primary subtypes [4,9]. IL-6 is known as a marker of iCAF and αSMA is a marker of myCAF. Our findings indicate that AREG may induce MSCs to differentiate into the iCAF phenotype. Furthermore, immunohistochemical analysis of human ESCC tissues revealed a significant correlation between high AREG expression and FAP, consistent with our in vitro findings demonstrating that AREG induces FAP expression. While our analysis of human ESCC tissue samples revealed no significant association between high AREG expression and prognosis, bioinformatics analysis indicated a significant upregulation of AREG expression in ESCC tissues compared with normal tissues. This finding suggests AREG upregulation might occur earlier in carcinogenesis rather than during later stages of ESCC progression. Previous studies have linked AREG with the activation of regulatory T cells that accumulate around tumor cells at the time of tumor development, suggesting a role for AREG in carcinogenesis [49,50]. Another study showed that AREG expression in the prostate gland gradually increases from benign to malignant stages, suggesting that AREG may contribute to the development of prostate adenocarcinoma [51]. These findings support our hypothesis that AREG contributes to the early stages of cancer development.

Our study has several limitations. First, the number of resected human ESCC samples used for clinical analysis in this study was relatively small. Increasing the sample size may reveal an association between high AREG expression and ESCC prognosis. Second, the role of EGFR in AREG-induced CAF differentiation and enhanced migration ability remains unclear. While we confirmed the expression of EGFR in MSCs and CAFs, pEGFR (Tyr1068) was not detected (Appendix A), suggesting that alternative phosphorylation sites or other pathways may be involved in mediating the effects of AREG on MSCs. Furthermore, although several phosphorylation sites other than Tyr1068 have been identified in EGFR, the phosphorylation status of these other sites was not analyzed in this study. Third, in vivo experiments were not conducted to validate the in vitro data. Previous studies have shown that AREG is crucial for ovarian cancer cell proliferation and metastasis in mice co-injected with CAFs and ovarian cancer cells [47]. Additionally, injection of AREG into mice transplanted with pancreatic cancer cells promoted tumor growth and metastasis [52]. Future studies using animal models are needed further to elucidate the role of AREG in ESCC progression.

## 5. Conclusions

This study demonstrates that direct contact between MSCs and ESCC cells enhances AREG secretion from ESCC cells, promoting their survival and migration through activation of the EGFR-Erk/p38 MAPK signaling pathway. Additionally, AREG promotes MSC migration and differentiation into CAFs. A schematic summary of the study results is presented in Figure 6. This study contributes to a deeper understanding of the complex interactions between cancer cells and the tumor microenvironment, specifically the role of CAFs in tumor progression. Our findings suggest that AREG could be a novel therapeutic target for ESCC.

## Figures and Tables

**Figure 1 cells-13-01733-f001:**
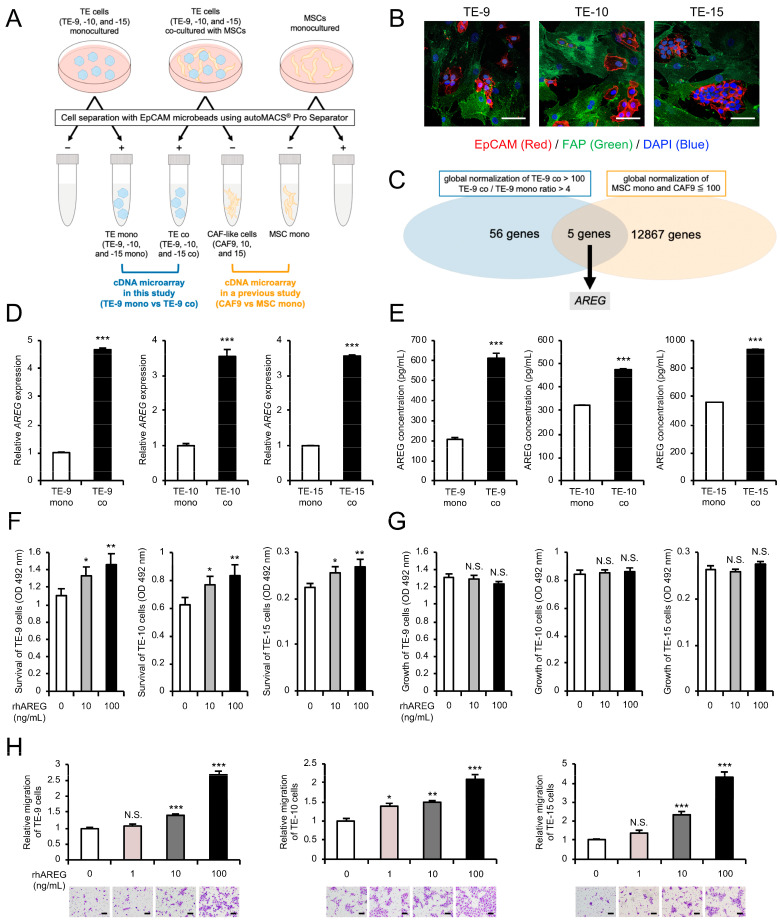
Amphiregulin (AREG) in esophageal squamous cell carcinoma (ESCC) cells induced by direct co-culture with cancer-associated fibroblast (CAF)-like cells promotes survival and migration of ESCC cells: (**A**) A schematic representation of the experimental design of the direct co-culture and cDNA microarray analysis. ESCC cell lines (TE-9, -10, and -15) and mesenchymal stem cells (MSCs) were co-cultured in the same dish for 4 d. Individually cultured ESCC cell lines and MSCs were prepared as control monocultures. Following monoculture or co-culture, cells were separated using epithelial cell adhesion molecule (EpCAM) microbeads. EpCAM-positive and EpCAM-negative cells after co-culture were defined as TE co (TE-9, -10, and -15 co) and CAF-like cells (CAF9, 10, and 15), respectively. Similarly, ESCC cell lines and MSCs after monoculture were defined as TE mono (TE-9, -10, and -15 mono) and MSC mono, respectively. While a previous study analyzed gene expression between MSC mono and CAF9, this study focuses on gene expression changes between TE-9 mono and TE-9 co using cDNA microarray analysis. (**B**) Double immunofluorescence staining for EpCAM (red) and FAP (green) was performed on a direct co-culture of ESCC cells and MSCs. The nucleus of each cell was counterstained with DAPI (blue). (**C**) Venn diagram depicting the overlap between genes exhibiting a global normalization threshold of TE-9 co > 100 and TE-9 co/TE-9 mono ratio > 4 in the cDNA microarray analysis as well as genes displaying a global normalization threshold of MSC mono and CAF9 < 100 in the previous analysis. Five genes were identified that overlapped between the two groups, with AREG showing the highest fold change. (**D**,**E**) The mRNA expression and secreted protein levels of AREG in TE mono and TE co were compared using qRT-PCR (**D**) and enzyme-linked immunosorbent assay (**E**). (**F**,**G**) The effects of recombinant human AREG (rhAREG) (10 and 100 ng/mL) on the survival (**F**) and growth (**G**) of ESCC cells were evaluated using the MTS assay. (**H**) The effect of rhAREG (1, 10, and 100 ng/mL) on the migration of ESCC cells was evaluated using the transwell migration assay. ESCC cells were seeded in the upper chamber, and migrated cells were counted in five representative fields of view using a microscope after 48 h. Representative images for each condition are presented below the graphs. The data are presented as the mean ± standard error of the mean (SEM) of three independent experiments (**D**–**H**). N.S., not significant; * *p* < 0.05, ** *p* < 0.01, *** *p* < 0.001. Scale bars: 50 μm (**B**); and 100 μm (**H**).

**Figure 2 cells-13-01733-f002:**
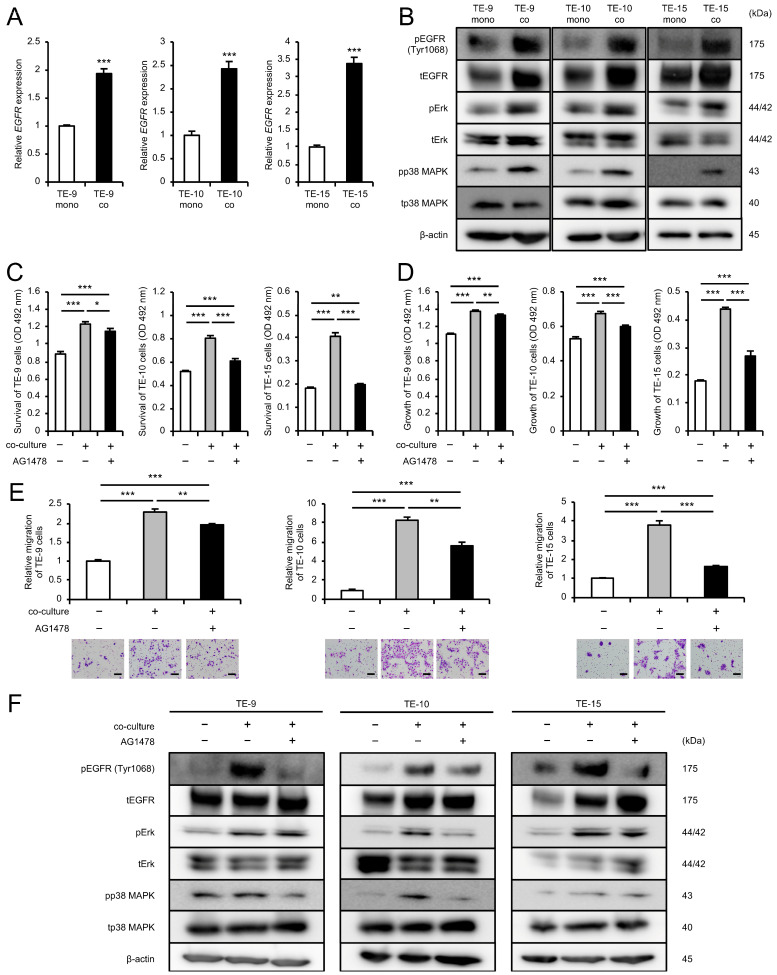
Direct co-culture with CAF-like cells promotes survival, growth, and migration of ESCC cells through the activation of the EGFR-Erk/p38 MAPK signaling pathway: (**A**) The mRNA expression levels of *EGFR* in TE mono and TE co were compared using qRT-PCR. (**B**) The protein expression levels of EGFR, pEGFR (Tyr1068), Erk, pErk, p38 MAPK, and pp38 MAPK in TE mono and TE co were compared using Western blotting. β-actin was used as a loading control. (**C**–**E**) To investigate the role of EGFR phosphorylation on the survival, growth, and migration of ESCC cells co-cultured with CAF-like cells, the effect of EGFR tyrosine kinase inhibitor AG1478 was examined in using the MTS assay (**C**,**D**) and transwell migration assay (**E**). ESCC cells were pretreated with AG1478 (10 μM) or DMSO as a control for 24 h before the start of the co-culture. ESCC cells were seeded in the upper chamber, and migrated cells were counted in five fields of view after 48 h (**E**). Representative images for each condition are presented below the graphs (**E**). (**F**) To investigate the effect of AG1478 or DMSO on the signaling pathways activated in ESCC cells co-cultured with CAFs, the protein expression levels of EGFR, pEGFR (Tyr1068), Erk, pErk, p38 MAPK, and pp38 MAPK in TE mono and TE co were compared using Western blotting. ESCC cells were treated with AG1478 (10 μM) or DMSO as a control for 24 h before co-culture. β-actin was used as a loading control. The data are presented as the mean ± SEM of three independent experiments (**A**,**C**–**E**). * *p* < 0.05, ** *p* < 0.01, *** *p* < 0.001. Scale bars: 100 μm (**E**).

**Figure 3 cells-13-01733-f003:**
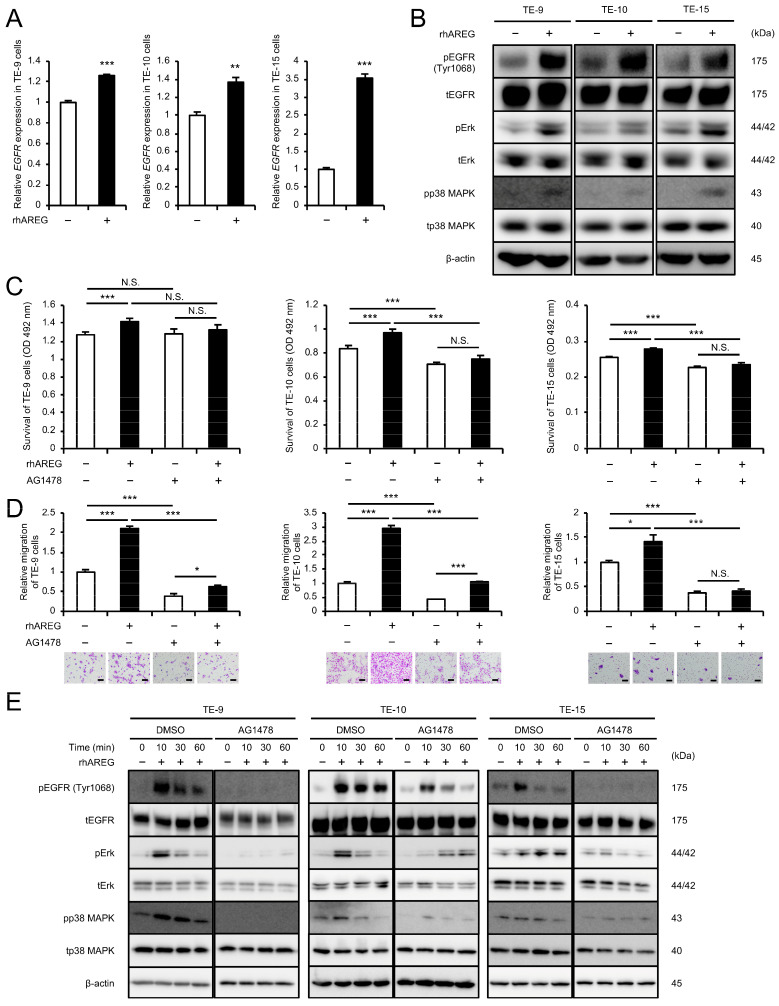
AREG promotes survival and migration of ESCC cells via the EGFR-Erk/p38 MAPK signaling pathway: (**A**) To investigate the effect of AREG on the mRNA expression levels of *EGFR* in ESCC cells, qRT-PCR was performed in TE cells with or without rhAREG (100 ng/mL). (**B**) To investigate the effect of AREG on the protein expression levels of EGFR, pEGFR (Tyr1068), Erk, pErk, p38 MAPK, and pp38 MAPK in ESCC cells, Western blotting was performed in TE cells with or without rhAREG (100 ng/mL). (**C**,**D**) To investigate the effects of AG1478 on the rhAREG-induced survival and migration of ESCC cells, the MTS assay (**C**) and transwell migration assay (**D**) were performed. ESCC cells were pretreated with AG1478 (10 μM) or DMSO as a control for 24 h before each assay. ESCC cells were seeded in the upper chamber, and migrated cells were counted in five fields of view after 48 h (**D**). Representative images for each condition are presented below the graphs (**D**). (**E**) To investigate the effect of AG1478 on rhAREG-induced signaling pathways in ESCC cells, the protein expression levels of EGFR, pEGFR (Tyr1068), Erk, pErk, p38 MAPK, and pp38 MAPK were compared using Western blotting. ESCC cells were pretreated with AG1478 (10 μM) or DMSO as a control for 24 h before treatment with rhAREG for 0, 10, 30, and 60 min. β-actin was used as a loading control. The data are presented as the mean ± SEM of three independent experiments (**A**,**C**,**D**). N.S., not significant; * *p* < 0.05, ** *p* < 0.01, *** *p* < 0.001. Scale bars: 100 μm (**D**).

**Figure 4 cells-13-01733-f004:**
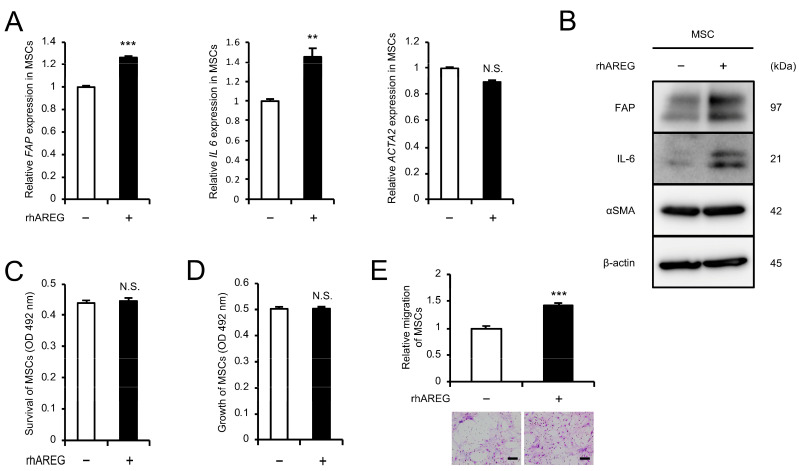
AREG promotes migration and CAF-like differentiation of MSCs: (**A**,**B**) To investigate the differentiation of MSCs into CAFs upon treatment with rhAREG (100 ng/mL), the mRNA and protein expression levels of FAP, IL-6, and αSMA were compared using qRT-PCR (**A**) and Western blotting (**B**). β-actin was used as a loading control (**B**). (**C**,**D**) The effects of rhAREG (100 ng/mL) on the survival (**C**) and growth (**D**) of MSCs were evaluated using the MTS assay. (**E**) The effect of rhAREG (100 ng/mL) on the migration of MSCs was evaluated using the transwell migration assay. MSCs were seeded in the upper chamber, and migrated cells were counted in five fields of view after 48 h. Representative images for each condition are presented below the graph (**E**). The data are presented as the mean ± SEM of three independent experiments (**A**,**C**–**E**). N.S., not significant; ** *p* < 0.01, *** *p* < 0.001. Scale bars: 100 μm (**E**).

**Figure 5 cells-13-01733-f005:**
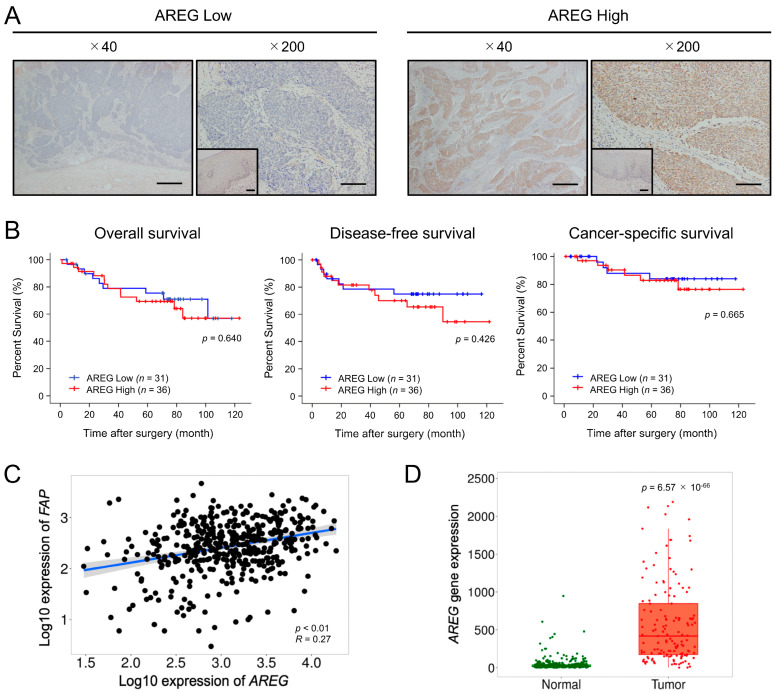
Significance of AREG expression in ESCC tissues: (**A**) Representative images of low (left) and high (right) expression of AREG in the tumor-invasive front (low-power field, 40×; high-power field, 200×). The corresponding normal squamous epithelium is shown as an inset within the high-power field. Scale bars: 400 μm (low-power field and insets in high-power field); and 100 μm (high-power field). (**B**) Kaplan–Meier analysis of overall survival, disease-free survival, and cancer-specific survival in 67 patients (one patient was excluded from the analysis due to a lack of postoperative outcomes) with ESCC stratified by AREG immunohistochemical staining intensity. The data were analyzed using the log-rank test. (**C**) Spearman correlation analysis of *AREG* and *FAP* gene expression levels in ESCC tissues using the TNMplot database. (**D**) *AREG* gene expression levels in normal and ESCC tissues using the TNMplot database.

**Figure 6 cells-13-01733-f006:**
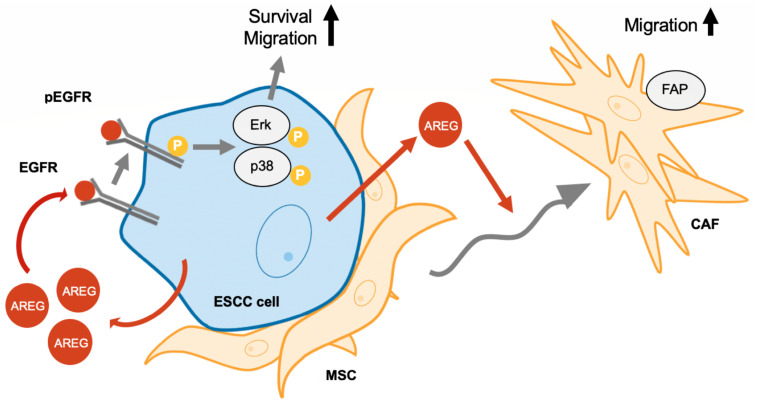
A schematic representation of AREG’s role in ESCC-MSC interactions. AREG secretion from ESCC cells, enhanced through direct contact with MSCs, promotes cell survival and migration via the EGFR-Erk/p38 MAPK signaling pathway. AREG also enhances the migration of MSCs and their differentiation into CAFs.

**Table 1 cells-13-01733-t001:** Genes markedly upregulated in TE-9 co compared to TE-9 mono but with low expression in MSC mono and CAF9.

Accession Number	Symbol	Description	Global Normalization				Ratio
			MSC mono	CAF9	TE-9 mono	TE-9 co	TE-9 co/TE-9 mono
NM_001657.3	*AREG*	amphiregulin		30	143	1224	8.54
NM_004833.1	*AIM2*	absent in melanoma 2		23	64	498	7.76
NM_004833.1	*AIM2*	absent in melanoma 2	2	35	81	541	6.70
NM_014398.3	*LAMP3*	lysosomal associated membrane protein 3	11	20	48	311	6.41
NM_006536.5	*CLCA2*	chloride channel accessory 2	14	41	49	311	6.33

According to DNA sequences at the National Center for Biotechnology Information “https://www.ncbi.nlm.nih.gov (accessed on 15 August 2024)”. MSC mono, mesenchymal stem cell monoculture; CAF, cancer-associated fibroblast; TE-9 mono, TE-9 monoculture; TE-9 co, TE-9 co-culture.

**Table 2 cells-13-01733-t002:** Association between AREG expression and clinicopathological factors of patients with ESCC.

		Expression of AREG		
	Case Number	Low (*n* = 31)	High (*n* = 37)	*p* Value
Age (years)				
<65	32	16	16	0.491
≥65	36	15	21	
Sex				
Male	14	7	7	0.710
Female	54	24	30	
Histological grade ^a^				
HGIEN + WDSCC	15	6	9	0.623
MDSCC + PDSCC	53	25	28	
Depth of tumor invasion ^a^				
T1	47	23	24	0.407
T2 + T3	21	8	13	
Lymphatic vessel invasion ^a^				
Negative	36	19	17	0.207
Positive	32	12	20	
Blood vessel invasion ^a^				
Negative	42	19	23	0.941
Positive	26	12	14	
Lymph node metastasis ^a^				
Negative	42	20	22	0.669
Positive	26	11	15	
Stage ^b^				
0 + I	37	17	20	0.948
II + III + IV	31	14	17	
Expression of αSMA ^c^				
Low	35	19	16	0.138
High	33	12	21	
Expression of FAP ^c^				
Low	38	23	15	0.005 *
High	30	8	22	

Data were assessed using the Chi-squared test. *p* < 0.05 was considered statistically significant: * *p* < 0.05. ^a^ Based on the 10th edition of the Japanese Classification of Esophageal Cancer [18,19]: HGIEN, high-grade intraepithelial neoplasia; WDSCC, well-differentiated squamous cell carcinoma; MDSCC, moderately differentiated squamous cell carcinoma; PDSCC, poorly differentiated squamous cell carcinoma. T1, tumor invades the mucosa and submucosa; T2, tumor invades the muscularis propria; T3, tumor invades the adventitia. ^b^ Based on the 7th edition of TNM classification by the Union for International Cancer Control [20]. ^c^ Patients were classified into low and high groups based on the immunoreactivity at the tumor-invasive front. The cutoff value was set at 30% (high: above 30%, low: below 30%) [10].

## Data Availability

The data presented in this study are available on request from the corresponding author.

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
