# Peer review of "AREG Upregulation in Cancer Cells via Direct Interaction with Cancer-Associated Fibroblasts Promotes Esophageal Squamous Cell Carcinoma Progression Through EGFR-Erk/p38 MAPK Signaling"

_cells, 2024, doi:10.3390/cells13201733_

Round 1

Reviewer 1 Report

Comments and Suggestions for Authors

The manuscript by Nakanishi et al. explores the role of secreted amphiregulin (AREG) in the physiology of esophageal squamous cell carcinoma (ESCC). The authors found that CAFs promote the expression and secretion of AREG by ESCC cells, thereby enhancing their survival and migration via the EGFR-Erk/p38 MAPK signaling pathway. Conversely, secreted AREG induces the expression of the CAF marker FAP. Through bioinformatics analysis, the authors confirmed a correlation between upregulated AREG in ESCC and the pathological characteristics of ESCC patients, suggesting that AREG could be a novel therapeutic target for ESCC. While the study presents interesting and novel data on targeting AREG in ESCC, there are some limitations in the experimental design that require major revision before publication.

  1. Please include controls to confirm that TE cells express EpCAM, as not all cancer cells exhibit this marker.
  2. Please provide a comment on why EGFR was not identified in the RNAseq analysis of TE cells.
  3. Please indicate the positions of protein markers on the Western blot images to ensure clarity in the presentation of results.
  4. Demonstrate the effect of recombinant AREG on the polarization of mesenchymal stem cells (MSCs) into myofibroblastic or inflammatory phenotypes. Using only one marker, FAP, is insufficient to draw a strong conclusion regarding this important aspect of CAFs polarization.
  5. Expand an introduction of AREG in introduction or result sections for better understanding of a protein.

Reviewer 2 Report

Comments and Suggestions for Authors

This manuscript by Nakanishi et al. explores the role of AREG upregulation in the progression of ESCC through its interaction with cancer-associated fibroblasts. The study offers valuable insights into how AREG influences ESCC cell survival, migration, and CAF differentiation via the EGFR-Erk/p38 MAPK pathway. The experimental results are compelling, and the findings are both novel and strongly supported by the presented data. I recommend accepting the manuscript in its current form.

Reviewer 3 Report

Comments and Suggestions for Authors

The authors establish a co-culture of ESCC cells and MSCs in the present manuscript. In such a way, through different approaches, they show a marked upregulation of amphiregulin  (AREG) in co-cultured ESCC cells and, thus, investigate the role of AREG in ESCC progression and its effects on CAFs.

The manuscripts reporting in the same context the co-presence of tumour cells and their related microenvironment are intriguing. The paper is well-written and the figures are of good quality. There are a lot of papers talking about the fact that cancerous cells release soluble factors targeting CAFs and not all are cited. However, it can be understandable, given the large amount of papers published. In addition, recent papers indicate that EGF (autocrine released) by colon cancer cells can induce the phosphorylation of EGFR on Tyr 1068 and, thus, the activation of MAPK, supporting in part the data of the authors. 

However, I have different concerns:

1. The authors should show the photographs of the co-cultures (figure 1)

2. in material and methods the authors should explain better which is the origin of CAFs. Do they derive from other cell types? How...? In the results section, concerning CAFs, they write about CAFs 10 or 15 for example. They should add a table or details about the cells and specify better.

3. The ESCC cells employed, three human ESCC cell lines (TE-9, -10, and -15) are different from any points of view?

4. On line 321, the authors write "We focused on EGFR and phosphorylated EGFR (pEGFR) 321 (Tyr1068), a reported marker of EGFR activation". However, it is not the only site of EGFR phosphorylation. The authors should explain better.

5. In addition to EGFR inhibitor have the authors used an EGF neutralizing antibody? Is there an autocrine or paracrine release of EGF?

6. Fig 1B, please improve beta-actin.

7. In Figs 1C and 1D there are discrepancies in survìvival and growth among the different cell lines, also considering the efficacy of EGFR inhibitor. Why? have the authors data that can explain these findings?

8. However the data in fig 6B seem not to be significant. 

9. siRNA approaches should be considered to verify the final hypothesis of the authors.

10. The final model is very good, but open a questions. AREG also enhances the migration of MSCs and their differentiation into CAFs. How does it occurr?

Comments on the Quality of English Language

It is quite fine

Round 2

Reviewer 1 Report

Comments and Suggestions for Authors

The manuscript has been improved, and it can now be accepted for publication.

Reviewer 3 Report

Comments and Suggestions for Authors

It is improved. Check, please only the figures and their resolution. 

Comments on the Quality of English Language

ok